# *Toxoplasma gondii* Rhoptry Protein 7 (ROP7) Interacts with NLRP3 and Promotes Inflammasome Hyperactivation in THP-1-Derived Macrophages

**DOI:** 10.3390/cells11101630

**Published:** 2022-05-12

**Authors:** Lijun Zhu, Wanjun Qi, Guang Yang, Yurong Yang, Yuwen Wang, Lulu Zheng, Yongfeng Fu, Xunjia Cheng

**Affiliations:** 1Department of Medical Microbiology and Parasitology, School of Basic Medical Sciences, Fudan University, Shanghai 200032, China; 14301010045@fudan.edu.cn (L.Z.); 15301010056@fudan.edu.cn (W.Q.); xjcheng@shmu.edu.cn (X.C.); 2Department of Pathogen Biology, School of Medicine, Jinan University, Guangzhou 510632, China; tyangguang@jnu.edu.cn; 3College of Veterinary Medicine, Henan Agricultural University, Zhengzhou 450046, China; yangyu7712@sina.com; 4Engineering Research Center of Optical Instrument and System, The Ministry of Education, Shanghai Key Laboratory of Modern Optical System, University of Shanghai for Science and Technology, Shanghai 200093, China; yuwenwang1995@outlook.com (Y.W.); llzheng@usst.edu.cn (L.Z.)

**Keywords:** *Toxoplasma gondii*, NLRP3 inflammasome, rhoptry protein, inflammasome hyperactivation

## Abstract

*Toxoplasma gondii* is a common opportunistic protozoan pathogen that can parasitize the karyocytes of humans and virtually all other warm-blooded animals. In the host’s innate immune response to *T. gondii* infection, inflammasomes can mediate the maturation of pro-IL-1β and pro-IL-18, which further enhances the immune response. However, how intercellular parasites specifically provoke inflammasome activation remains unclear. In this study, we found that the T. gondii secretory protein, rhoptry protein 7 (ROP7), could interact with the NACHT domain of NLRP3 through liquid chromatography-mass spectrometry analysis and co-immunoprecipitation assays. When expressing ROP7 in differentiated THP-1 cells, there was significant up-regulation in NF-κB and continuous release of IL-1β. This process is pyroptosis-independent and leads to inflammasome hyperactivation through the IL-1β/NF-κB/NLRP3 feedback loop. The loss of ROP7 in tachyzoites did not affect parasite proliferation in host cells but did attenuate parasite-induced inflammatory activity. In conclusion, these findings unveil that a *T. gondii*-derived protein is able to promote inflammasome activation, and further study of ROP7 will deepen our understanding of host innate immunity to parasites.

## 1. Introduction

*Toxoplasma gondii* is an obligate intracellular parasite that has the ability to parasitize virtually all karyocytes of humans and other warm-blooded animals. *T. gondii* is a global health hazard with a seroprevalence ranging from single digits to above 60% in different regions [1]. In healthy individuals, *T. gondii* infection does not usually develop overt symptoms because of a fully functioning immune system, but it can result in toxoplasmosis or abortion in immunocompromised individuals or pregnant women, conditions that can be life-threatening [2].

The innate immune system includes NK cells, neutrophils, dendritic cells, monocytes, and macrophages as the first line of defense against *T. gondii* invasion [3]. In innate immunity, cytosolic protein complexes called inflammasomes have recently been shown to participate in host resistance to *T. gondii* [4,5]. Once activated, inflammasomes produce the active form of the proinflammatory cytokines IL-1β and IL-18 [6], both of which provide protection against *T. gondii* infection [7,8]. Among all types of inflammasomes, the NLRP3 inflammasome has been the most extensively studied and plays a crucial role in *T. gondii* resistance in humans and mice, as shown by the uncontrolled growth of and increased susceptibility to the parasite after NLRP3-inflammasome-related gene knockouts [5,9,10].

Inflammasome activation is a complicated response. The canonical activation of NLRP3 inflammasomes requires two signals [11]. The first is the priming signal, which is derived from certain pathogen-associated molecular patterns (PAMPs) or damage-associated molecular patterns. The priming signal leads to NF-κB activation and upregulates the production of inflammasome proteins and cytokines, along with many posttranslational modifications [12], such as ubiquitination, deubiquitination, phosphorylation, and dephosphorylation. The second is the activating signal caused by a variety of stimuli, which results in intracellular ion concentration changes, mitochondrial dysfunction, lysosomal disruption, and inflammasome assembly. Moreover, there is the caspase-4/5/11-dependent non-canonical pathway and the alternative pathway [13]. Given that macrophages infected with type II *T. gondii* exhibit a significant increase in inflammatory cytokine secretion [5], type II *T. gondii* appears to provide both signals 1 and 2 for NLRP3 inflammasome activation. Previous studies have demonstrated that GRA15 from type II strains, a *T. gondii* secretion protein from dense granules, serves as the signal that activates NF-κB signaling through TNF receptor-associated factors [10,14]. A study reported that phosphorylation GRA7 could interact with the PYD domain of ASC to facilitate ASC oligomerization and inflammasome activation [15]. Though parasite proteins may be related to inflammasome activation, little is known about how *T. gondii* activates the NLRP3 inflammasome.

A rhoptry is a specialized secretory organelle in apicomplexans, including *T. gondii*, that secretes a group of functional proteins called rhoptry proteins (ROPs), which are mainly discharged during *T. gondii* invasion. After secretion, some ROPs are located within the parasitophorous vacuole or at the parasitophorous vacuole membrane (PVM), whereas others enter the host-cell cytosol or nucleus [16]. A special class of ROPs form the rhoptry protein kinase (ROPK) family and are characterized by a conserved kinase domain [17]. It has been recently demonstrated that some ROPKs, such as ROP5, ROP16, and ROP18, can regulate their host innate immunity [18,19]. However, the function of most ROPKs remains unclear.

Recently, some studies have shown that endogenous or exogenous proteins interacting with NLRP3 enable the inflammasome to license its activation [19,20,21,22]. Here, this report identified that a known rhoptry kinase family protein, ROP7, had the ability to bind with the NACHT domain of NLRP3 through co-immunoprecipitation assays and mass spectrometry analysis. ROP7 expression in THP-1 cells promoted NF-κB up-regulation and induced pyroptosis-independent inflammasome activation. Loss of ROP7 in tachyzoites did not affect parasite proliferation in host cells but did attenuate parasite-induced inflammatory activity. Furthermore, we demonstrated the role of the IL-1β/NF-κB/NLRP3 loop in ROP7- and parasite-induced inflammasome activation. Taken together, these findings indicate that ROP7 is a novel effector promoting NLRP3 inflammasome activation during *T. gondii* infection. Further study of ROP7 will deepen our understanding of host innate immunity to parasites.

## 2. Materials and Methods

### 2.1. Cell Lines and Parasite Strains

Human embryonic kidney cell line 293T cells, human leukemia monocytic cell line THP-1 cells, and human foreskin fibroblasts (HFF) cells were obtained from the Cell Bank of the Chinese Academy of Sciences (Shanghai, China). The 293T and HFF cells were cultured in DMEM. The THP-1 cells were cultured in RPMI1640. All media were supplemented with 10% fetal bovine serum, and for tetracycline-regulated THP-1 cells, tet-free FBS was used. The cells were maintained at 37 °C with 5% CO_2_.

The *T. gondii* ME49 strain was grown as a tachyzoite in HFF cells as described previously [23].

### 2.2. Reagents and Antibodies

Antibodies: anti-IL-1β (NB600-633, Novus Biologicals, Littleton, CO, USA); anti-HA (Ab137838, Abcam, Cambridge, UK); anti-Flag (F1804, Sigma-Aldrich, St Louis, MO, USA); anti-NLRP3 (AG-20B-0014-C100, Adipogen, San Diego, CA, USA); anti-ASC (AG-25B-0006-C100, Adipogen, San Diego, CA, USA)

Reagents: recombinant human IL-1β (IL038, Sigma-Aldrich, St Louis, MO, USA), recombinant human TNF-α (ab9642, Abcam, Cambridge, UK); Z-YVAD-FMK (S8507, Selleck, Shanghai, China); Z-VAD-FMK (S7023, Selleck, Shanghai, China); MCC950 (S8930, Selleck, Shanghai, China); Bay 11-7082 (S2913, Selleck, Shanghai, China); IL-1R antagonist (sc-221747, Santa Cruz Biotechnology, Santa Cruz, CA, USA).

### 2.3. Plasmid Construction

FLAG-HA-pcDNA3.1- (#52535, Addgene, Cambridge, MA, USA), pcDNA3-HA, pcDNA3-C-EGFP, pCMV-MCS-FLAG, pCMV-tag2B, pmCherry, and pLVX-TetOne-Puro were used to construct expression plasmids. All the primers used in this study are listed in Appendix A. The *Rop7* (XM_018782264) DNA template was derived from *T. gondii* ME49 genomic DNA using a commercial DNA extraction kit (69506, QIAGEN, Hilden, Germany). The pro-IL-1β and pro-caspase-1 cDNA templates were derived from Raw264.7 RNA after cDNA synthesis using an RNA extraction kit (74136, QIAGEN, Hilden, Germany) and a cDNA synthesis kit (6210A, Takara Bio, Dalian, Liaoning, China). PCR was performed using Platinum *Taq* DNA Polymerase High Fidelity (11304011, Invitrogen, Carlsbad, CA, USA). Cloning was carried out using the NovoRec plus One-step PCR Cloning Kit (NR005-01A, Novoprotein, Suzhou, Jiangsu, China).

To construct tetracycline-inducible ROP7 expression plasmids, *Rop7* was inserted into pLVX-TetOne-Puro using the NovoRec plus One-step PCR Cloning Kit.

### 2.4. Co-Immunoprecipitation Assays and Liquid Chromatography/Mass Spectrometry (LC/MS) Screening

To detect protein interaction, THP-1 or 293T transfected with the corresponding plasmids were collected and lysed in RIPA Lysis Buffer. The lysates were centrifuged, and 10% of the total resulting supernatant was retained as input. The supernatants were then incubated at 4 °C with the corresponding primary antibody and rProtein G beads (SA016005, Smart-Lifesciences, Changzhou, Jiangsu, China) for every 2 h. The beads were washed with RIPA Lysis Buffer three times and then mixed with sample buffer. After denaturing and SDS-PAGE, the gel was subjected to liquid chromatography/mass spectrometry (LC/MS) analysis or Western blot.

LC/MS raw data were further identified by Sequest and Proteome Discoverer (Thermo Scientific, Waltham, MA, USA) using a *T. gondii* database from Uniprot.

Raw MS data files are available from public repositories MassIVE (MassIVE ID MSV000088602; https://massive.ucsd.edu/ProteoSAFe/dataset.jsp?accession=MSV000088602, accessed on 7 May 2022).

### 2.5. Western Blotting

After lysis, the protein samples were degenerated and separated by SDS-PAGE. Then, the gels were transferred onto polyvinylidene difluoride (PVDF) membranes. After blocking with 10% skim milk, the proteins on PVDF membranes were incubated with specific primary antibodies overnight at 4 °C. HRP-conjugated secondary antibodies were used for the corresponding primary antibodies and visualized by enhanced chemiluminescence reagents.

### 2.6. Enzyme-Linked Immunosorbent Assay (ELISA)

ELISA was used to detect the level of IL-1β and TNF-α in cell-free medium collected from 293T or THP-1 cells. Cell-free medium was centrifuged at 200× *g* at 4 °C to remove residual cells, and the level of mouse IL-1β, human IL-1β, and human TNF-α were measured using the mouse IL-1β ELISA kit (88-8013, Invitrogen, Carlsbad, CA, USA), human IL-1β ELISA kit (DY201-05, R&D Systems, Minneapolis, MN, USA), or human TNF-α ELISA kit, (DY210-05, R&D Systems, Minneapolis, MN, USA) respectively, according to the manufacturer’s protocols.

### 2.7. NLRP3 Inflammasome Reconstitution in 293T Cells

NLRP3 inflammasomes were reconstituted in 293T cells to explore the ROP7 functioning.

To analyze IL-1β secretion, the 293T cells were seeded in 24-well plates and co-transfected with 15-ng pcDNA3-N-Flag-NLRP3 (#75127, Addgene, Cambridge, MA, USA), 5-ng pcDNA3-N-Flag-ASC1 (#75134, Addgene, Cambridge, MA, USA), 100-ng pCMV-pro-IL-1β, and 2.5-ng pCMV-pro-caspase-1-FLAG through Lipofectamine 3000 (L3000015, Invitrogen, Carlsbad, CA, USA). For the analysis of IL-1β cleavage, the 293T cells were seeded in six-well plates and co-transfected with 150-ng pcDNA3-N-Flag-NLRP3, 50-ng pcDNA3-N-Flag-ASC1, 1000-ng pCMV-pro-IL-1β, and 25-ng pCMV-pro-caspase-1-FLAG. Different doses of pcDNA3.1-HA-ROP7 or empty vectors were transfected simultaneously. Cells and cell-free medium were collected 36 h after transfection. IL-1β secretion and cleavage were measured via ELISA and Western blot.

### 2.8. Confocal Microscope Analysis of Protein Co-Localization

The 293T cells grown on glass-bottom cell culture dishes were transfected with pcDNA3-NLRP3-EGFP, and pmCherry or pmCherry-ROP7. Protein co-localization was captured via a confocal microscopy (LSM900, Zeiss, Oberkochen, Germany). Imaging and image processing were performed using Zeiss ZEN 3.1.

### 2.9. Generation of Human THP-1 Cell Lines Expressing ROP7 Stably

The generation of human THP-1_ROP7_ cell lines through lentiviral transduction followed a previous study [24]. Briefly, for the lentivirus package, a total of 24-μg plasmids (pLVX-Tet-ROP7, psPAX2, and pMD2.G at a molar ratio of 1:1:1) were transfected into 293T cells. After 2- and 3-days post-infection, cell culture supernatant was harvested and stored at −80 °C. For lentiviral transduction, lentivirus was added to THP-1 cells overnight with 8-μM polybrene at different concentrations. One day later, the infected cells were selected with 1-μg/mL puromycin.

### 2.10. LDH Assay

Fresh cell-free supernatants were collected following centrifugation and temporarily stored at 4 °C. The corresponding cells were lysed completely by the lysis buffer supplied in the LDH Detection Kit as maximum enzyme activity controls. The LDH release levels were determined according to the manufacturer’s protocols. The absorbance was read at 440 nm, and the mean values were normalized using the maximum enzyme activity.

### 2.11. qPCR

qPCR was performed to detect the transcription of genes associated with the NF-κB pathway (*IL1B* (NM_000576.3), *TNFA* (NM_000594.4), *NLRP3* (NM_001243133.2), *NFKBIA* (NM_020529.3)). *ACTB* (actin beta, NM_001101.5) was used as a housekeeping gene. Total RNA was extracted by RNeasy Plus Mini Kit (74134, QIAGEN, Hilden, Germany), and cDNA was synthesized by PrimeScript™ II 1st Strand cDNA Synthesis Kit (6210A, Takara Bio, Dalian, Liaoning, China). After cDNA synthesis, the reaction products were mixed with TB Green^®^ Premix Ex Taq™ II (RR820A, Takara Bio, Dalian, Liaoning, China) and analyzed by ABI 7500 (Applied Biosystems, Foster City, CA, USA). mRNA fold changes were calculated using the 2^−ΔΔCq^ method.

### 2.12. NF-κB Dual-Luciferase Reporter Gene Assay

The 293T cells in the 48-well plate were co-transfected with 200-ng NF-κB-Luc Reporter plasmids and pRL-TK at a molar ratio between 10:1 and 200-ng pcDNA3-HA-ROP7 or pcDNA3-HA. Twenty-four hours post-transfection, 10 ng/mL of TNF-α was added to 293T as a positive control. Forty-eight hours post-transfection, luciferase activity was measured using the Dual-Luciferase Reporter Gene Assay Kit (11402ES60, Yeasen, Shanghai, China) and GloMax^®^ 20/20 Luminometer (Promega, Madison, WI, USA) according to the manufacturer’s protocol.

### 2.13. T. gondii ME49 ROP7 Knockout

The CRISPR/Cas9 plasmids, pUPRT::DHFR-D (#58528, Addgene, Cambridge, MA, USA) and pSAG1::CAS9-U6::sgUPRT (#54467, Addgene, Cambridge, MA, USA), were generously given by David Sibley. The ROP7 gRNA sequence was designed according to the Eukaryotic Pathogen CRISPR guide RNA/DNA Design Tool [25]. The PCR primers for plasmid mutagenesis and homologous recombination are listed in Appendix A.

The gene knockout and complement of *T. gondii* ME49 followed Shen B. et al. [26] and Sidik et al. [27]. Briefly, pSAG1::CAS9-U6::sgROP7 and pROP7::DHFR-D were constructed using Q5 High-Fidelity DNA Polymerase (#E0555L, NEB, Ipswich, MA, USA) and the NovoRec plus One-step PCR Cloning Kit. The plasmids (between 80-μg pSAG1::CAS9-U6::sgROP7 and 20-μg pROP7::DHFR-D) were transfected into 10^6^ tachyzoites in a final volume of 400-μL Cytomix Buffer through the Gene Pulser Xcell Eukaryotic System (#1652661, Bio-Rad, Hercules, CA, USA). The electroporation parameters were 1.7 kV for two 176-μs pulses in 5-s intervals in 0.4-cm cuvettes. After electroporation, tachyzoites were inoculated into HFF cells, and 4-µM pyrimethamine was used to select the positive clone.

### 2.14. Plaque Assays

For plaque assays, the HFF cells were grown to confluent monolayers in six-well plates and infected with 1000 Δ*Rop7* or WT *T. gondii* tachyzoites. At 5 days post-infection, the plaques formed by the growing parasites were counted under a microscope.

### 2.15. THP-1 Infection with T. gondii

For better effect and infection consistency, tachyzoites were inoculated into THP-1 according to the following steps unless otherwise noted.

The THP-1 cells were inoculated at 3.75 × 10^5^/mL. After PMA differentiation, the THP-1 cells were infected with tachyzoites at MOI = 1. At 2 h post-infection, the cells were washed with PBS and placed in fresh medium. At 6 h post-infection, the cells were washed again. The supernatant and cells were harvested at different times.

### 2.16. Statistical Analysis

All data were analyzed using GraphPad Prism 7. For all calculations, *p*-values < 0.05 were considered statistically significant.

## 3. Results

### 3.1. T. gondii Secretory Protein ROP7 Is Associated with NLRP3 and Promotes the Maturation of IL-1β in 293T Cells

To explore the relationship between NLRP3 inflammasomes and T. gondii proteins, a co-immunoprecipitation screening assay was conducted on T. gondii-ME49-strain-infected 293T cells overexpressing NLRP3. After protein mass spectrometry analysis, a total of 333 T. gondii-related proteins from 293T-NLRP3 cells were identified (Figure 1A). To identify appropriate candidate proteins, we first excluded those in which only a single peptide was detected. Then, given that NLRP3 inflammasomes occur in the cytoplasm, we further narrowed the range of potential candidates to secretory proteins, which resulted in a final list of 20 proteins (Appendix A). Further screening was performed using co-immunoprecipitation assays by co-transfecting candidate proteins with NLRP3 into 293T cells and evaluating the effect of the IL-1β secretion by the 293T-reconstituted NLRP3 inflammasomes system.

The microneme protein MIC3 had the highest score of Sequest HT among all the screened secreted proteins, and its interaction with NLRP3 was confirmed in a Co-IP assay (Appendix A). However, its biological function in the regulation of NLRP3 could not be speculated on as there was no significant change in the IL-1β secretion in 293T-reconstituted NLRP3 inflammasomes compared with the empty vector control (Figure 1B). The rhoptry protein 7 (ROP7) was our next target. Similar to MIC3, ROP7 also interacted with NLRP3 in a Co-IP assay in 293T cells (Figure 1C). However, contrary to MIC3, when the empty vector was replaced with ROP7 in 293T-reconstituted NLRP3 inflammasomes, the IL-1β in the supernatant was much higher in the resulting supernatant (Figure 1B), and pro-IL-1β cleavage was enhanced in cell lysates (Figure 1D). This process was dependent on the existence of caspase-1, as the significant reduction in IL-1β secretion without caspase-1 transfection (Appendix A). In addition, the co-localization analysis conducted on 293T cells via confocal microscopy revealed that NLRP3 localized with ROP7 (Figure 1E and Appendix A). Therefore, ROP7 could be associated with NLRP3 and may promote IL-1β maturation.

### 3.2. NLRP3 and ROP7 Interact through the NACHT Domain and ROP7_AA35-255_

ROP7 belongs to the ROP2 family or ROPKs family, which possesses characteristic kinase-like folds. However, the kinase-like domain of ROP7 is generally regarded as a pseudokinase as it does not possess a key catalytic loop. To investigate whether ROP7 can in fact catalyze and how it interacts with NLRP3, we constructed a series of expression plasmids containing the NLRP3 or ROP7 domains. The intracellular sensor NLRP3 consists of three domains, namely, PYD, NACHT, and LRR, whereas ROP7 is divided into a nonkinase region (ROP7_AA35-255_) and a pseudokinase domain (ROP7_AA256-543_; Figure 2A). To investigate which domains interacted with ROP7, plasmids expressing PYD, NACHT, or LRR were co-transfected into 293T cells with ROP7. Co-immunoprecipitation revealed that ROP7 interacted with NLRP3 through the NACHT domain but not the PYD or the LRR domains (Figure 2B). Next, express plasmids containing ROP7_AA35-255_ and ROP7_AA256-543_ were co-transfected into 293T cells with NACHT to investigate whether the ROP7 pseudokinase domain had actual activity participating in inflammasome activation. Unexpectedly, ROP7_AA256-543_ from the pseudokinase domain did not precipitate with NACHT. On the contrary, ROP7_AA35-255_ may have been associated with NACHT (Figure 2C). These results were further confirmed in reconstituted NLRP3 inflammasomes; moreover, ROP7_AA35~255_ alone was sufficient to promote IL-1β secretion, whereas ROP7 _AA256-543_ was dispensable (Figure 2D).

Taken together, our results indicated that the nonkinase region of ROP7 can interact with the NACHT domain of NLRP3 and promote IL-1β maturation.

### 3.3. ROP7 Expression in THP-1 Induces NLRP3 Inflammasome Hyperactivation

Based on the results above with 293T, we hypothesized that ROP7 could also affect natural NLRP3 inflammasome activation in macrophages. To test this hypothesis, we constructed a tetracycline-regulated ROP7 expression system in a THP-1 cell line (THP-1_ROP7_), which could express ROP7 in the presence of doxycycline (Appendix A). Similar to previous results, NLRP3 also co-immunoprecipitated with ROP7 in THP-1 after doxycycline treatment (Figure 3A). In an NLRP3 inflammasome complex, the NLRP3 protein interacts with ASC via the PYD domain. Therefore, ASC should precipitate with NLRP3 if ROP7 interacts with NLRP3. In fact, we observed that ASC could be detected in THP-1_ROP7_ co-immunoprecipitation samples (Figure 3A) but not in 293T co-expression samples (Appendix A), indicating that ROP7 pulled down ASC through its interaction with NLRP3.

We next determined whether ROP7 influenced natural inflammasome activation. After PMA differentiation, continuous secretion of IL-1β and TNF-α, more expression of pro-IL-1β, and greater cleavage of pro-IL-1β in THP-1_ROP7_ cells compared with THP-1_Mock_ cells were observed (Figure 3B,C). We also observed a much higher IL-1β secretion in THP-1_ROP7_ following nigericin treatment (Figure 3D). It should be noted that continuous secretion only occurred after THP-1 differentiation, although the induction of doxycycline was slightly increased compared with that of non-doxycycline cells in a monocyte state (Appendix A). When the THP-1_ROP7_ cells were pretreated with an NLRP3-specific inhibitor (MCC950), caspase-1-specific inhibitor (Z-YVAD-FMK), or pan-caspase inhibitor (Z-VAD-FMK), the IL-1β and TNF-α secretion were significantly attenuated (Figure 3E), indicating that this phenomenon was inflammasome-dependent.

Generally, the release of IL-1β is accompanied by cell swelling and rupturing of the plasma membrane, which is known as pyroptosis [28]. However, several studies have reported that some inflammasome ligands, such as OatA-deficient *Staphylococcus aureus* [29] and *T. gondii* [30], can mediate IL-1β secretion from living macrophages, which is known as “hyperactivation.” In fact, THP-1_ROP7_ normally survived similarly to the wild type, which was indicative of a hyperactive status. To further verify this, LDH release in THP-1 was detected. After doxycycline induction, PMA differentiation, and LPS treatment, no difference was observed in the LDH release, whereas nigericin stimulation facilitated substantial LDH release and even higher LDH release in THP-1_ROP7_ (Figure 3F). Thus, ROP7 expression in THP-1 can facilitate IL-1β release and lead to macrophage hyperactivity.

### 3.4. ROP7 in THP-1 Triggers the Priming Signal but Not Derived from ROP7 Itself to NF-κB Pathway Directly

The canonical NLRP3 inflammasome activation needs both “priming” and “activating” signals. Our results have so far indicated that ROP7 expression triggers the entire activation cascade. Generally, the priming signal is provided by PAMP and leads to the activation of NF-κB, which then upregulates the transcription of pro-IL-1β and NLRP3 [31]. To further determine whether ROP7 expression induces transcription through the NF-κB signaling pathway, qPCR was employed to detect the transcription of genes associated with the NF-κB pathway, such as *IL1B*, *TNFA*, *NLRP3*, and *NFKBIA* (nuclear factor of κ-light polypeptide gene enhancer in B-cell inhibitor-α). The mRNA fold changes suggested NF-κB activation in THP-1_ROP7_ but none in the THP-1_Mock_ (Figure 4A). The results also indicated that these changes could be counteracted by pretreating cells with BAY 11-7082 (Figure 4B) and an NF-κB pathway inhibitor [32]. Notably, ROP7-induced activation was milder than LPS-treated activation.

Given that THP-1 cells are inflammatory cells that possess numerous elaborate mechanisms for regulating an inflammatory response, and to determine whether ROP7 is a direct effector that facilitates NF-κB activity, we performed an NF-κB reporter dual-luciferase assay in 293T by co-transfecting ROP7. Nevertheless, ROP7 overexpression in cells did not increase luciferase activity (Figure 4C), suggesting that ROP7 might not directly affect the NF-κB pathway.

It was abnormal that in Figure 3D, both NLRP3 and the caspase inhibitors attenuated IL-1β and TNF-α secretion in THP-1_ROP7_ when treated with doxycycline for 24 h, as these inhibitors should not decrease TNF-α secretion [33,34]. However, these results were further confirmed via qPCR (Figure 4D). The transcription of *NFKBIA* was significantly attenuated when the cells were pretreated with inflammasome inhibitors, indicating downregulation of NF-κB activity [35], though *IL1B*, *TNFA*, and *NLRP3* transcription were not as significant. It was also confirmed that NLRP3 or caspase inhibitors disrupted IL-1β secretion in LPS-primed or nigericin-stimulated THP-1_ROP7_ cells but did not disrupt TNF-α secretion (Figure 4E). Thus, the priming step is likely derived from other pathways in which ROP7-induced inflammasome activation is involved.

Together, these results indicate that THP-1_ROP7_ possesses the priming signal but that the signal is not directly derived from ROP7 itself to the NF-κB pathway directly, indicating that other mechanisms must be involved during THP-1_ROP7_ priming.

### 3.5. The IL-1β/NF-κB/NLRP3-Positive Feedback Loop Participates in ROP7-Induced Inflammasome Activation

IL-1β, a proinflammatory factor, plays a significant role in the inflammatory response. Recently, some studies have demonstrated that the IL-1β/NF-κB/NLRP3-positive feedback loop participates in some pathophysiologic processes [36,37]. Thus, we inferred that this positive loop might at least partially engage in inflammasome activation in THP-1_ROP7_. At first, the priming effects of IL-1β in differentiated THP-1 were tested using recombinant proteins. After stimulating recombinant IL-1β, the transcription of *IL1B*, *TNFA*, *NLRP3*, and *NFKBIA* was upregulated (Figure 5A). Next, an IL-1β receptor antagonist (IL-1Ra, CAS 566914-00-9) was used to verify whether this loop participated in the ROP7-induced inflammasome activation. IL-1Ra is a specific inhibitor of the TIR domain-mediated MyD88/IL1-RI interaction and does not disrupt the LPS-induced TLR4/MyD88 interaction [38]. When THP-1_ROP7_ was pretreated with IL-1Ra, a clear attenuation in both IL-1β and TNF-α secretion in the resulting supernatant was observed (Figure 5B). The qPCR results confirmed that IL-1Ra significantly inhibited the NF-κB activity in THP-1_ROP7_ (Figure 5C). However, none of the results indicated that IL-1Ra influenced LPS and nigericin functioning. Therefore, the inflammasome activation in THP-1_ROP7_ is dependent on the IL-1β/NF-κB/NLRP3-positive feedback loop, at least to a certain extent.

### 3.6. ROP7 and the IL-1β/NF-κB/NLRP3 Loop Are Involved in Parasite-Induced Inflammasome Response in Macrophages

To further elucidate the role of ROP7 in parasite infection, a ROP7-knockout ME49 strain (Δ*Rop7*) was constructed using CRISPR/CAS9 plasmids. Previous studies have demonstrated that ROP7 is not a virulent protein and that ROP7 knockout does not affect cyst number in mouse brains over long periods of time [39]. Similar to previous reports, no difference was observed in invasion and proliferation between Δ*Rop7* and WT strains, as plaque counts in HFF cells were equivalent (Figure 6A). Previous studies have reported that several common *T. gondii* strains can activate inflammasomes and promote cytokine secretion in macrophages. Here, infection with the wild-type ME49 strain promoted IL-1β in THP-1 cells. However, for the Δ*Rop7* strain, IL-1β secretion significantly declined within 24 h, at which time parasites continued to proliferate but did not egress from parasitophorous vacuoles (Figure 6B). In cell lysates, pro-IL-1β in WT- and Δ*Rop7*-infected THP-1 cells also revealed an attenuation in inflammatory response (Figure 6C). Since the *Sag*1 transcription levels did not change (Appendix A), this attenuation was not due to the alteration of the parasite growth rate. Accordingly, these results confirmed the integral role of ROP7 in inflammasome response.

Next, we explored whether the IL-1β/NF-κB/NLRP3 loop could influence parasite-induced inflammasome activation. At first, there was no difference in IL-1β secretion between THP-1 and IL-1Ra-pretreated THP-1 cells (MOI = 1; see Appendix A). However, within 12 h after MOI increased from 1 to 5, a reduction in IL-1β secretion was observed in the IL-1Ra-pretreated group (Figure 3D). This reduction disappeared again after 18 h. Therefore, the IL-1β/NF-κB/NLRP3 loop influences parasite-induced inflammasome activation, at least to a certain extent.

Above all, our results indicate that both ROP7 and the IL-1β/NF-κB/NLRP3 loop are critical for parasite-induced inflammasome response.

## 4. Discussion

The innate immune system is the host’s first line of defense against *T. gondii*. A well-established mouse model exists for understanding the role of IL-12 and IFN-γ in the host’s innate immune response to *T. gondii* [3]. IL-12, produced by dendritic cells, neutrophils, and monocytes, induces natural killer (NK) cells to produce IFN-γ. Subsequently, IFN-γ induces the expression of a specific set of genes called IFN-γ-inducible genes, which include immunity-related GTPases and guanylate-binding proteins. IFN-γ can additionally trigger a robust adaptive immune response. During this cascade of events, the production of inflammasomes, IL-1β and IL-18, facilitates host innate immunity in restricting *T. gondii* invasion by synergizing with or replacing IL-12 in activating NK cells and triggering IFN-γ secretion [7,40].

Discovered in 2002 [41], inflammasomes are known to participate in various anti-infection and pathophysiologic responses [42,43]. *T. gondii* has been known to activate NLRP1 and NLRP3 inflammasomes [5,9], and the loss of these complexes will result in increased parasite proliferation. In human or murine macrophages, typical NLRP3 inflammasome activation requires two signals, namely, priming signal 1 for transcription and activating signal 2 for activated complexes. During this process, endogenous or exogenous proteins interacting with NLRP3 enable the inflammasome to license its activation. For example, the interaction between NLRP3 and mitochondrial antiviral signaling protein (MAVS) promotes NLRP3 recruitment to mitochondria [20], while stress granule protein DDX3X [44] and some virus proteins [21,22] can facilitate NLRP3 assembly through interacting with the NACHT or LRR domain. Previous studies have demonstrated that *T. gondii* type II strains possess both of these signals. For signal 1, GRA15 from type II strains can activate the NF-κB pathway by interacting with TRAFs in macrophages [14,45]. Type II strains can also activate the Syk-CARD9-NF-κB axis to release IL-1β in human monocytes [30]. Nevertheless, it was unknown which *T. gondii* effector provided signal 2 for NLRP3 or NLRP1 activation, despite the fact that most strains promote the secretion of mature IL-1β regardless of the genotype or virulence [5]. In this report, using a Co-IP assay and LC/MS screening, we found that ROP7 from the ME49 strain interacts with NLRP3. Expressing ROP7 in THP-1 cells results in a sustained release of IL-1β without pyroptosis and also in the production of more IL-1β than control cells that undergo subsequent LPS stimulation or nigericin activation.

Although ROP7 has been known for nearly 20 years, its function remains uncharacterized, similar to other *T. gondii*-secreted proteins [46,47]. ROP7 is not a virulence factor, and loss of ROP7 does not affect parasite invasion, proliferation, or egression [39]. ROP7 is most commonly researched in the context of a vaccine [48] owing to its conserved gene sequence [49] across *T. gondii* strains and CD8^+^ T cell epitopes [50]. Here we found that ROP7 knockout in tachyzoites reduced the inflammatory response of THP-1-derived macrophages. THP-1 cells are human leukemia monocytic cell lines that have similar patterns as do human peripheral blood mononuclear cell-derived macrophages after PMA differentiating [51]. It was revealed that ROP7 possesses a role in host immunity to parasites. ROP7 belongs to the ROP2 family, which is characterized by a kinase-like fold [52]. However, half the members of this family, including ROP7, lack a key catalytic loop and are thus not expected to possess the enzymatic activity and are considered pseudokinases. In fact, our results indicate that the putative kinase-like domain of ROP7 is not the key structure for NLRP3 activation but that the nonkinase region is indispensable. Normally, ROP7 is stored in *T. gondii* rhoptries and co-localizes with ROP1 [47]. During the cellular invasion, ROP7 is released into the host-cell cytoplasm or translocated to the PVM, which has the potential to directly connect to NLRP3 [53].

We further demonstrated that the nonkinase region of ROP7 interacts with the NLRP3 NACHT domain to stimulate the secretion of IL-1β. NLRP3 consists of three domains, namely, PYD, LRR, and NACHT. PYD is responsible for interacting with the adaptor protein ASC, whereas the LRR and NACHT domains bind to NEK7 and are involved in an active conformational change of NLRP3 [54]. In fact, NACHT alone can be fully activated by NLRP3 inflammasomes [55], and the NLRP3 inhibitor, MCC950, can target this domain to induce conformational changes that inactivate NLRP3 [56,57]. We also showed that MCC950 can significantly inhibit ROP7-mediated IL-1β secretion. Based on these observations, we deduced that ROP7 likely affects a conformational change in NLRP3 that transitions it from an inactive to an active state. Similar mechanisms have been reported in the Zika NS5 protein and the EV71 3D protein [21,22]. Another probable mechanism is that ROP7 targets the post-translational modifications of NLRP3 such as HUWE1 [58] or protein kinase D [59]. So further studies are necessary to unveil how ROP7 induces NLRP3 inflammasome activation. In addition, NLRP1, another inflammasome that responds to *T. gondii* infection, also possesses a NACHT domain. It is noteworthy that *T. gondii* induces NLRP1-dependent pyroptosis in the macrophages of parasite-resistant rats and that this response provides Lewis rats and other outbred rats with resistance to parasite dissemination [60]. Whether ROP7 can bind with NLRP1 and activate an NLRP1 inflammasome is not known.

Previous studies show that significant IL-1β secretion without pyroptosis occurs in *T. gondii*-infected human or mouse macrophages [5]. Importantly, we demonstrated that the expression of ROP7 in differentiated-THP-1 cells leads to continuous IL-1β secretion without pyroptosis, which is consistent with the characteristics of *T. gondii* infection. The cellular state that is characterized by normal IL-1β secretion without pyroptosis is termed “hyperactivation.” Besides *T. gondii*, other pathogens or stimuli, such as OatA-deficient *Staphylococcus aureus* and oxidized lipids, can also induce IL-1β release without pyroptosis [61]. The specific mechanisms that determine whether cells undergo pyroptosis or hyperactivation are still debated. Here, our results indicate that ROP7 may be a new hyperactivation ligand. Hyperactivation is considered to be beneficial for CD4^+^ T cell response but will ignite many immune-related diseases if it is uncontrolled [62]. During *T. gondii* infection, several studies have demonstrated that IL-1β enhances local or systemic symptoms in mouse models. IL-1β and IL-1 signaling can induce ileitis during acute infection through peroral infection and cachexia during chronic infection in mice [63,64]. Furthermore, IL-1β may also be a critical factor for parasite antagonization of host–antiparasitic response [65]. Conversely, the survival of hyperactivated cells may contribute to parasite dissemination because circulating immune cells are the most important carriers for parasites to migrate through tissues and organs [66,67]. Therefore, whether parasite-induced hyperactivation is involved in secondary disease, immune evasion, or parasite dissemination needs further exploration.

ROP7-induced inflammasome hyperactivation depends on the IL-1β/NF-κB/NLRP3-positive loop. Initially, we observed a significant increase in NF-κB-related mRNA transcription in THP-1_ROP7_ cells, suggesting that ROP7 might have a priming function. However, ROP7 could not upregulate NF-κB activity directly, and we instead investigated the positive feedback loop. The IL-1β/NF-κB/NLRP3-positive loop, or IL-1β autocrine loop, participates in the pathogenesis of many inflammatory diseases. Through IL-1R, IL-1β can trigger a signaling cascade that includes MYD88, IRAKs, TRAF6, and NF-κB and results in the transcription of pro-IL-1β and other proinflammatory cytokines. pro-IL-1β is cleaved following inflammasome activation, which forms a positive feedback loop [68]. As there is no pyroptosis in parasite-infected macrophages, it is reasonable to consider that positive feedback may provide an actuating signal for persistent cell hyperactivation. When the THP-1_ROP7_ cells were pretreated with an IL-1R antagonist, the NF-κB and inflammasome activity in THP-1_ROP7_ decreased. This indicates what is required of this positive feedback for continuous hyperactivation. The positive feedback loop was also investigated during the parasite infection of THP-1. Given the activation intensity, we increased MOI from 1 to 5, which resulted in a slight decline in IL-1β secretion within 12 h after the IL-1Ra treatment. However, such a decline did not persist for more than 18 h. It may be attributed to the proliferative tachyzoites that result in more other effectors, such as GRA7, GRA14, or GRA15, which have the ability to activate NF-κB [69] and thereby compensate for any deficiency in IL-1R signaling.

ROP7 knockout can attenuate parasite-induced inflammasome activation but cannot nullify inflammasome activation. This is likely because parasite proteins other than ROP7 also participate in the promotion of NLRP3 activation. For example, phosphorylation of GRA7 is capable of regulating inflammasome oligomerization and activation [15]. GRA35, GRA42, and GRA43 are required for Lewis rat macrophages to undergo pyroptosis, although the underlying molecular mechanism has not been uncovered [70]. Parasite mitogen-activated protein kinases and the surface antigen 2A are also related to inflammasome activation [71]. GRA9 interacts with NLRP3 and disrupts NLRP3 inflammasome assembly [72]. Therefore, further study is required to understand whether and how these proteins synergize with each other to regulate the inflammasome activation pathway.

In conclusion, our findings associate ROP7 with inflammasome activation, a novel role for the protein. We have characterized and described the role of ROP7 in inflammasome activation, i.e., that ROP7 interacts with the NLRP3 NACHT domain and induces cell hyperactivation, a process that depends on an IL-1β/NF-κB/NLRP3-positive loop. Moreover, the proinflammatory role of ROP7 was demonstrated during the *T. gondii* infection of macrophages. These findings will further improve our understanding of host innate immunity to parasites.

## Figures and Tables

**Figure 1 cells-11-01630-f001:**
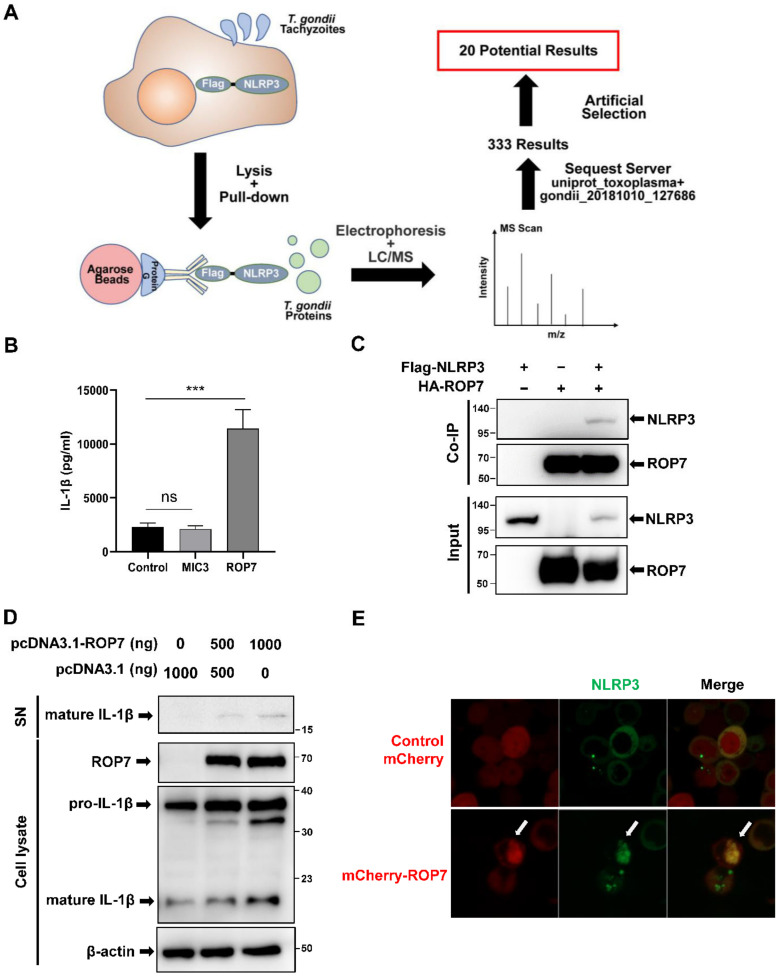
ROP7 is associated with NLRP3 and promotes the maturation of IL-1β in 293T cells. (**A**) Schematic for the identification of potential proteins interacting with NLRP3. Protein G binding with anti-Flag-precipitated parasite proteins from 293T cell lysates following parasite invasion. Immunoprecipitated proteins were separated by SDS-PAGE and further analyzed by LC/MS after elution. Parasite proteins were identified using a *T. gondii* database from Uniprot. (**B**) The 293T cells were co-transfected with inflammasome-component plasmids (pro-IL-1β, pro-caspase-1, NLRP3, and ASC) and ROP7, MIC3, or an empty vector. The IL-1β secretion in cell-free supernatant was measured after 36 h. Data are expressed as mean ± SEM values. ns: non-significant, *** *p* < 0.001, as compared with the empty vector groups. (**C**) The 293T cells were transfected with pcDNA3-HA-ROP7 and pcDNA3.1-Flag-NLRP3. The cell lysates were subjected to immunoprecipitation using anti-HA antibody and further analyzed by immunoblotting with an anti-Flag antibody (NLRP3) and an anti-HA antibody (ROP7). (**D**) The 293T cells were co-transfected with inflammasome-component plasmids and different concentrations of ROP7 or an empty vector. At 48 h post-transfection, the cells and cell-free medium were harvested respectively, and the cleavage of pro-IL-1β was detected by Western blot using an anti-Flag antibody. (**E**) The 293T cells were transfected with pcDNA3-NLRP3-EGFP and pmCherry or pmCherry-ROP7. At 48 h post-infection, ROP7 and NLRP3 co-localization was captured via confocal microscopy.

**Figure 2 cells-11-01630-f002:**
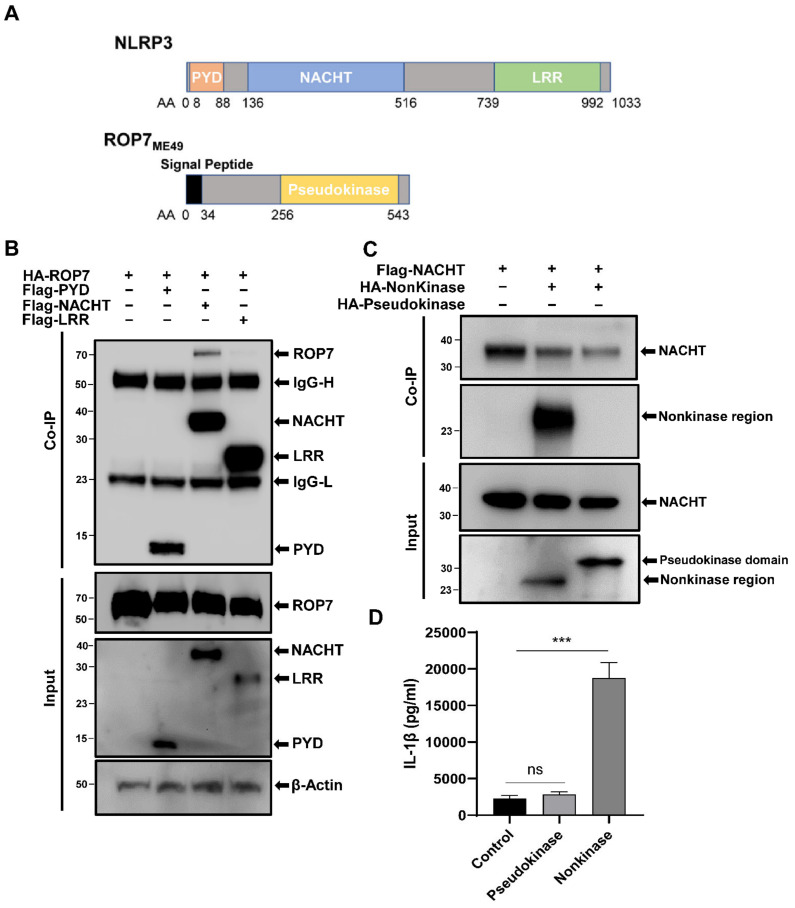
The interaction of NLRP3 and ROP7 domains in 293T cells. (**A**) Schematic structure of NLRP3 and ROP7. (**B**) The 293T cells were transfected with pcDNA3-HA-ROP7 in combination with pCMV-Tag2B-PYD, pCMV-Tag2B-NACHT, or pCMV-Tag2B-LRR. The cell lysates were immunoprecipitated using an anti-Flag antibody and further analyzed by immunoblotting using an anti-Flag antibody (PYD/NACHT/LRR) and an anti-HA antibody (ROP7). (**C**) The 293T cells were transfected with pCMV-Tag2B-NACHT in combination with pcDNA3-HA-nonkinase or pcDNA3-HA-Kinase. The cell lysates were immunoprecipitated using an anti-Flag antibody and further analyzed by immunoblotting using an anti-Flag antibody (NACHT) and an anti-HA antibody (nonkinase/pseudokinase). (**D**) The 293T cells were co-transfected with inflammasome-component plasmids and ROP7 pseudokinase domains, nonkinase regions, or empty plasmids. The IL-1β secretion in cell-free supernatant was measured after 36 h. Data are expressed as mean ± SEM values. ns: non-significant, *** *p* < 0.001, as compared with the empty vector groups.

**Figure 3 cells-11-01630-f003:**
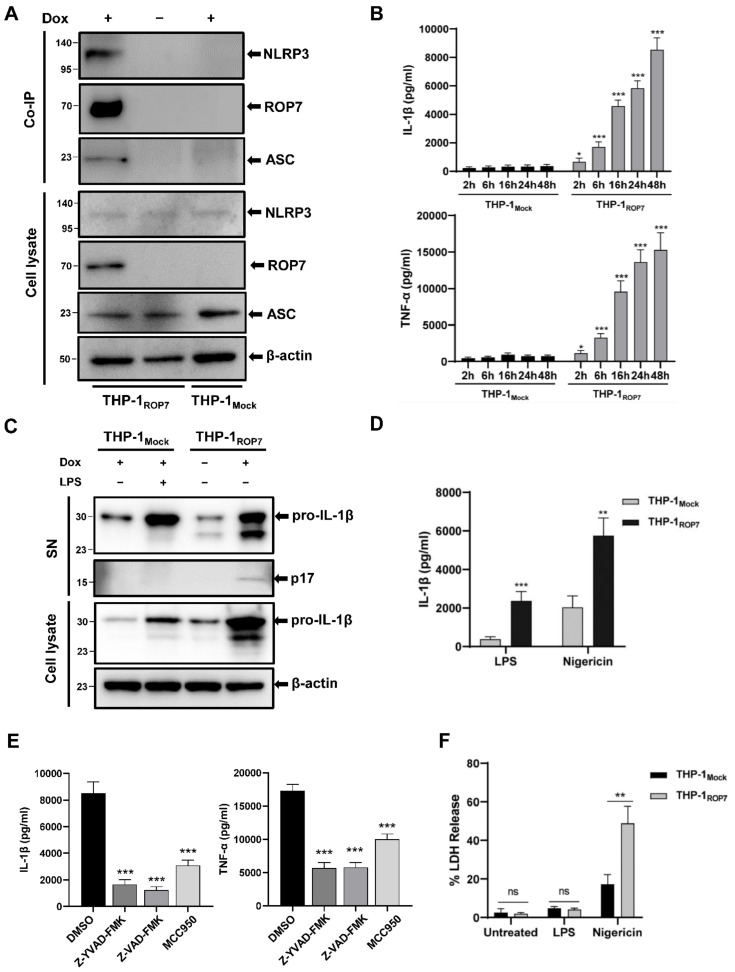
ROP7 in THP-1 cells activated NLRP3 inflammasome and induced a hyperactive status in cells. THP-1_Mock_ and THP-1_ROP7_ were induced by 1-μg/mL doxycycline for 48 h and differentiated under 100-nM PMA for another 48 h. THP-1 infected with lentivirus as a mock control, THP-1_Mock_. (**A**) NLRP3 and ASC were co-precipitated with Flag-HA-ROP7 in THP-1 cells. After expression and differentiation, the cells were harvested for Co-IP using an anti-HA antibody to immunoprecipitated Flag-HA-ROP7 and further analyzed by Western blot using an anti-NLRP3, anti-ASC, and anti-Flag antibody. (**B**) After differentiation, cell medium was renewed and collected periodically to detect IL-1β and TNF-α secretion by ELISA. Data are expressed as mean ± SEM values. * *p* < 0.05, *** *p* < 0.001, as compared with THP-1_Mock_. (**C**) pro-IL-1β and p17 of THP-1_Mock_ and THP-1_ROP7_ in cell lysate and supernatant (SN) were detected by immunoblotting using anti-IL-1β. In total, 100-ng/mL LPS was used in THP-1_Mock_ for 4 h as a positive control. (**D**) At 24 h post-induction, the cells were treated with 100-ng/mL LPS for 4 h alone or 10-μM nigericin for 1 h subsequently. Supernatants were collected for ELISA. Data are expressed as mean ± SEM values. ** *p* < 0.01, *** *p* < 0.001, as compared with THP-1_Mock_. (**E**) After PMA differentiation, 10-μM of Z-YVAD-FMK, Z-VAD-FMK, or MCC950 was added to the cells for 24 h. The IL-1β and TNF-α in cell-free supernatant were detected by ELISA. Data are expressed as mean ± SEM values. *** *p* < 0.001, as compared with the DMSO group. (**F**) Supernatants that were untreated (24 h after PMA differentiation and medium replacement), LPS-treated (100 ng/mL for 4 h), and nigericin-treated (10 μM for 1 h) were collected for the detection of LDH release. The results were normalized by the maximum enzyme activity in each group. Data are expressed as mean ± SEM values. ns: non-significant, ** *p* < 0.01.

**Figure 4 cells-11-01630-f004:**
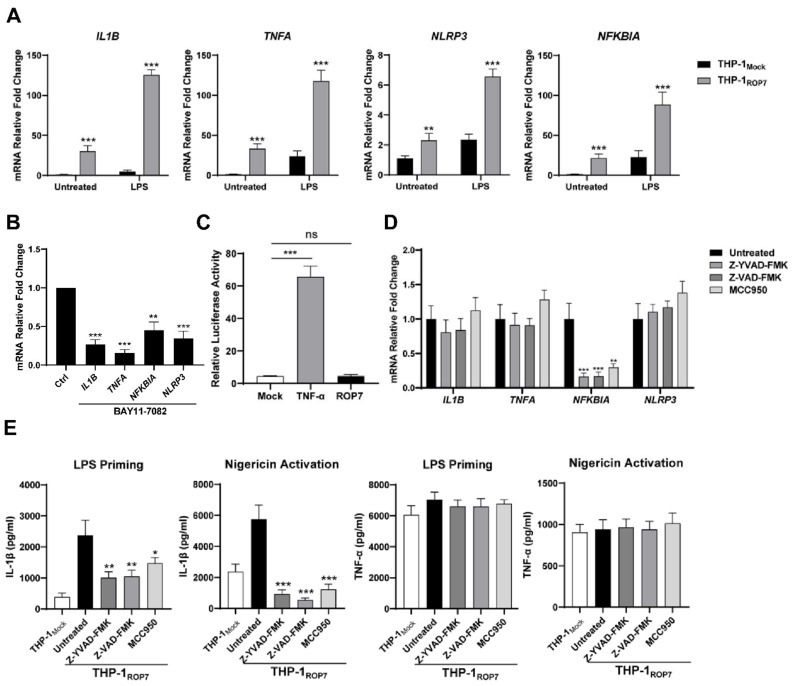
The NF-κB signaling pathway was activated in THP-1_ROP7_, but its activation was not directly derived from ROP7. THP-1_Mock_ and THP-1_ROP7_ were induced by 1-μg/mL doxycycline for 48 h and differentiated under 100-nM PMA for another 48 h (**A**) or with 10-μM BAY11-7082 in THP-1_ROP7_. (**B**) Total RNA was extracted, and the mRNA relative fold changes were determined via RT-qPCR. (**C**) The 293T cells were co-transfected with a dual-luciferase reporter plasmid (ranilla: firefly 1:10) for NF-κB. pcDNA3-HA-ROP7 or empty plasmids (mock). At 48 h post-transfection, luciferase activity was examined, and the values were normalized according to the ratio of firefly luciferase activity and ranilla luciferase activity. In total, 20-ng/mL TNF-α was used as a positive control. (**D**) THP-1_ROP7_ cells were induced by 1-μg/mL doxycycline for 48 h and differentiated under 100-nM PMA for another 48 h along with 10-μM Z-YVAD-FMK, Z-VAD-FMK, or MCC950. Total RNA was extracted, and the mRNA relative fold changes were determined via RT-qPCR. (**E**) THP-1_Mock_ and THP-1_ROP7_ were induced by 1-μg/mL doxycycline for 48 h and differentiated under 100-nM PMA for another 48 h. Then, 100-ng/mL LPS for 4 h and 10-μM nigericin for 1 h subsequently were used to activate NLRP3 inflammasomes. In total, 10-μM Z-YVAD-FMK, Z-VAD-FMK, or MCC950 was added with LPS or nigericin. The IL-1β and TNF-α supernatant concentrations were detected by ELISA. Data are expressed as mean ± SEM values. ns: non-significant, * *p* < 0.05, ** *p* < 0.01, *** *p* < 0.001, as compared with untreated groups.

**Figure 5 cells-11-01630-f005:**
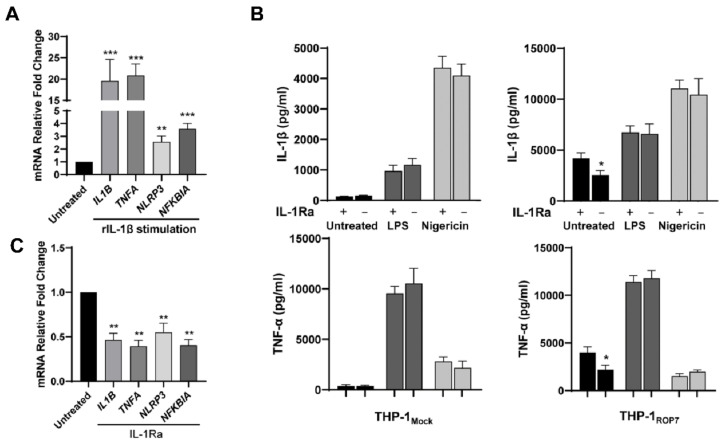
IL-1R antagonist reduces ROP7-induced IL-1β secretion. (**A**) THP-1 ROP7 were differentiated under 100-nM PMA and stimulated with 10-ng/mL rIL-1β. The mRNA relative fold changes were determined via RT-qPCR. Data are expressed as mean ± SEM values. ** *p* < 0.01, *** *p* < 0.001, as compared with untreated groups. (**B**) THP-1_Mock_ and THP-1_ROP7_ were induced with 1-μg/mL doxycycline for 48 h and differentiated under 100-nM PMA for another 48 h with 30-μM IL-1Ra or alone. After 48 h differentiation, cell medium was renewed with or without IL-1Ra, and the fresh cell-free medium was collected after 12 h as untreated groups. Then, 100-ng/mL LPS for 4 h and 10-μM nigericin for 1 h subsequently were used with 30-μM IL-1Ra or alone. Cell-free supernatant was collected for ELISA. Data are expressed as mean ± SEM values, * *p* < 0.05. (**C**) THP-1_ROP7_ was induced by 1-μg/mL doxycycline for 48 h and differentiated under 100-nM PMA for another 48 h with 30-μM IL-1Ra or alone (untreated). The cells were collected for qPCR to detect relative mRNA fold changes. Data are expressed as mean ± SEM values. ** *p* < 0.01, as compared with untreated groups.

**Figure 6 cells-11-01630-f006:**
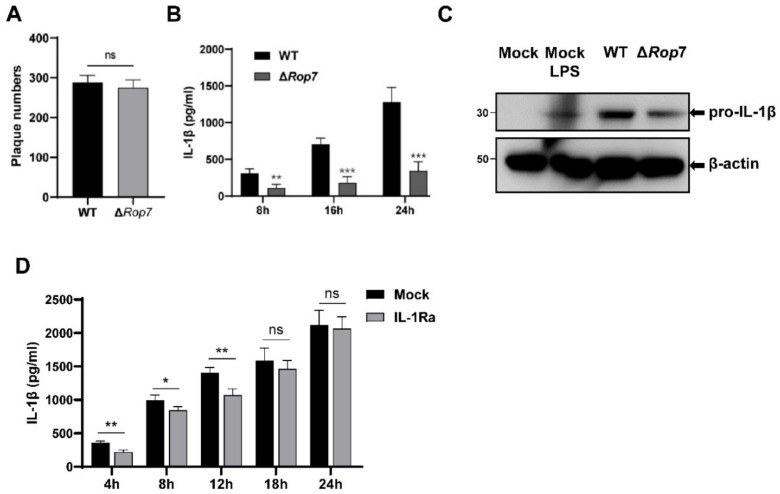
ROP7 knockout in T. gondii reduced parasite-induced IL-1β secretion in THP-1 cells. (**A**) Plaque assay of WT and Δ*Rop7* strains over 5 days infecting HFF. The plaques were counted using microscope. (**B**) After PMA differentiation, the THP-1 cells were infected with WT or Δ*Rop7* tachyzoites at MOI = 1. Supernatants were harvested at different time, and IL-1β was detected through ELISA. (**C**) The THP-1 cells were differentiated under 100-nM PMA for 48 h and infected with WT or Δ*Rop7* tachyzoites (MOI = 1). LPS-treated was a positive control. Pro-IL-1β cleavage was detected by immunoblotting using anti-IL-1β. (**D**) THP-1 were differentiated under 100-nM PMA for 48 h and treated with 30-μM IL-1Ra or nothing for 1 h before parasite invasion. The cells were infected with parasites (MOI = 5) with 30-μM IL-1Ra or none. The IL-1β in supernatants was detected by ELISA. Data are expressed as mean ± SEM values. ns: non-significant, * *p* < 0.05, ** *p* < 0.01, *** *p* < 0.001, as compared with mock groups.

## Data Availability

The data presented in this study are available on request from the corresponding authors.

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
