# Peer review of "Toxoplasma gondii Rhoptry Protein 7 (ROP7) Interacts with NLRP3 and Promotes Inflammasome Hyperactivation in THP-1-Derived Macrophages"

_cells, 2022, doi:10.3390/cells11101630_

Round 1
Reviewer 1 Report
The manuscript entitled “Toxoplasma gondii Rhoptry Protein 7 (ROP7) interacts with NLRP3 and promotes inflammasome hyperactivation in macrophages” I find very interesting and valuable.
Authors studied and described the role of ROP7 in activation of macrophages and in the inflammation process. They utilized several different laboratory methods to achieve interesting results; molecular methods, immunological methods, microscopy and cell culture.
In my opinion, descriptions in chapter Material and methods should be improved to become clearer. For example, the type of the cells used in the study should be presented first, not only the names of cell lines (293T, THP-1, HFF); the list of antibodies used in this study is presented as list of reagents that should be purchased; description concerning ELISA should be rewrite, it sounds not clear, when authors used the phrase “medium was measured” but they meant measurement of the level of IL-1β and TNF-α in the medium.
The results were described in details. The data obtained in this study were collected and presented in schemes and pictures which make them easy to read and analyse.
In the last chapter, Discussion, authors confronted results obtained in this study with data available in the literature.
The manuscript is well written and worth to be published. After minor corrections in chapter Materials and Methods I recommend this article to be published in Cell.
Author Response
We thank you for the positive evaluation of our work. Revision had been done according to the \comments and suggestions.

Reviewer 2 Report
The manuscript by Cheng et al reports an interesting finding that ROP7, a secreted protein from parasite Toxoplasma gondii, can directly interact with NLRP3 and modulate inflammasome activity in macrophage. The authors started with an immunoprecipitation-mass spectrometry analysis of NLRP3 interactome in T. gondii-infected HEK293T cells, specifically looking for T. gondii protein. After narrowing down the MS list to ROP7, they further verified and characterized its binding to NLRP3. The authors went on to explore the functional consequences of this interaction by using a ROP7 overexpressing THP-1 cell model and found that ROP7 induces macrophage hyperactivation in an uncanonical feed-forward fashion. Overall, this manuscript touches on an interesting topic and provides new perspectives with regard to inflammatory response to T. gondii infection. However, the authors still need to provide more evidence and clarification to support their primary hypothesis. The manuscript can be improved if the following specific points could be addressed:
- Were any controls included in the IP-MS analysis? Although ROP7 as an interaction partner of NRLP3 was confirmed by several different assays subsequently, the details of controls employed should be stated explicitly to support the quality of this experimental data.
- In figure 1E, the colocalization of ROP7 and NLRP3 was used to support the interaction between them. However, this was done in HEK293T cells in which both ROP7 and NLRP3 were overexpressed in the cytosol (which are diffusive in nature). From the images shown in the figure, it is difficult to tell if the interaction happens at a specific subcellular location. Also, it is obvious that the control mCherry shows significant colocalization with NLRP3 as well. The data will be more convincing if the author could show the colocalization of NLRP3 and ROP7 when both proteins, or at least one of them, are/is expressed at endogenous levels, i.e, NLRP3-expressing cells infected with gondii.
- Does the author have any other evidence showing inflammasome activation in ROP7-expressing cells beside IL-1beta secretion? Inflammasome-reconstituted HEK293T cell model was used to show that ROP7 induces IL-1beta secretion. Since both ASC and caspase-1 were co-transfected into the cell, were ASC specks observed in these cells? Did caspspase-1 get cleaved?
- The author used the ROP7 overexpressing THP-1 cells to characterize native NLRP3 inflammasome activation in macrophage. THP-1 cells were differentiated into macrophage by using PMA, which is a strong activator of the NF-kB pathway. This could potentially confound the interpretation of data especially when the authors hypothesize that ROP7 modulate NLRP3 via NF-kB pathway. The authors could try to use more canonically used cell models, i.e. bone marrow-derived macrophage or PBMC monocyte-derived macrophage in conjunction with the ROP7 knockout T. gondii mutant to dissect the function of ROP7 in NLRP3 inflammasome activation.
Round 2
Reviewer 2 Report
Thanks for the detailed response from the authors. The manuscript as improved significantly. But there are still a few points that needs clarification and key experiments should be performed to ensure the soundness of the overall conclusion.
Comment 2 and response 2:
If the authors think NLRP3 and ROP7 colocalize and forms aggregates, this should be highlighted in the figures with higher magnification. Also, the author should quantify how often these aggregates forms as compared to the control situation. Is it statistically different?
Comment 3 and response 3:
The author did not address whether ASC specks could be observed in their experiment. If not, why?
Comment 4 and response 4:
The author has already generated the ROP7 knockout T. gondii and used this strain in Figure 6. Therefore, performing experiments using bone marrow derived macrophage and the knockout strain should be feasible and will greatly support the overall conclusion of the manuscript.
